# Readiness of Mozambique Health Facilities to Address Undernutrition and Diarrhea in Children under Five: Indicators from 2018 and 2021 Survey Data

**DOI:** 10.3390/healthcare10071200

**Published:** 2022-06-27

**Authors:** Júlia Sambo, Adilson Fernando Loforte Bauhofer, Simone S. Boene, Marlene Djedje, António Júnior, Adalgisa Pilale, Luzia Gonçalves, Nilsa de Deus, Sérgio Chicumbe

**Affiliations:** 1Instituto Nacional de Saúde (INS), EN1, Bairro da Vila-Parcela nº 3943, Distrito de Marracuene, Maputo 1120, Mozambique; adilson.bauhofer@ins.gov.mz (A.F.L.B.); simonboene@gmail.com (S.S.B.); marlenedjedje1@gmail.com (M.D.); antonio.junior@ins.gov.mz (A.J.); nilsa.dedeus@ins.gov.mz (N.d.D.); sergio.chicumbe@ins.gov.mz (S.C.); 2Instituto de Higiene e Medicina Tropical (IHMT), Universidade Nova de Lisboa, 1349-008 Lisbon, Portugal; 3Pediatria, Hospital Central de Maputo (HCM), Maputo 164, Mozambique; adalgisapilale@gmail.com; 4Global Health and Tropical Medicine, Instituto de Higiene e Medicina Tropical, Universidade Nova de Lisboa, 1349-008 Lisbon, Portugal; luziag@ihmt.unl.pt; 5Departamento de Ciências Biológicas, Universidade Eduardo Mondlane (UEM), Maputo 1100, Mozambique

**Keywords:** health services, readiness, undernutrition, diarrhea, children, Mozambique

## Abstract

The World Health Organization’s systems framework shows that service delivery is key to addressing pressing health needs. Inadequate healthcare and the lack of healthcare services are factors associated with undernutrition and diarrhea in children under five, two health conditions with high morbi-mortality rates in Mozambique. The aim of the analysis was to determine the readiness score of nutrition and diarrhea services for children under five and the influence of malaria and HIV (Human Immunodeficiency Virus) service readiness on the readiness of these two services. A total of 1644 public health facilities in Mozambique were included from the 2018 Service Availability and Readiness Assessment. Additionally, a cross-sectional study was conducted to determine the availability and readiness scores of nutrition services in 2021 in five referral health facilities. The availability of nutrition and diarrhea services for children is low in Mozambique, with both scoring below 75%. Major unavailability was observed for human resources, guidelines, and training dimensions. Diarrhea (median (IQ): 72.2% (66.7 to 83.3)) and nutrition service readiness (median (IQ): 57.1% (52.4 to 57.1)) scores were significantly different (*p* < 0.001), while it is desirable for both services to be comprehensively ready. Nutrition services are positively associated with diarrhea service readiness and both services are associated with malaria and HIV service readiness (*p* < 0.05). None of the health facilities had all tracer items available and none of the facilities were considered ready (100%). There is a persisting need to invest comprehensively in readiness dimensions, within and across child health services.

## 1. Introduction

The morbi-mortality rate in children under five is a strong indicator of a country’s health condition, the quality of its health systems, and access to health services [1,2,3]. Although the globe recorded a decline in child mortality by almost half from 2000 to 2019, addressing the causes of poor child health and mortality remains a major challenge [4]. In low- and middle-income countries (LMIC), diseases such as undernutrition, diarrhea, malaria, and HIV present a high burden in children under five [4,5,6,7]. Undernutrition increases the risk of the severity of diarrhea and also the risk of death from diarrhea [8,9,10]. In 2011, two thirds of the deaths worldwide were attributed to diarrhea associated with undernutrition [8,9]. Specifically, diarrhea was considered the third leading cause of death in 2019, accounting for 9.9% of the deaths of children under five globally [4]. In 2018, 21.9% of children under five globally were estimated to be stunted and 7.3% were estimated to be wasted [11,12]. In Mozambique, in 2015, the prevalence of diarrhea in children under five was 11.0% and 41.7% were stunted, 4.4% were wasted, and 15.6% were underweight [13,14,15,16]. A Mozambican hospital-based cross-sectional study reported a frequency of 54.1% for undernutrition among children under five with diarrhea [17]. The child’s nutritional status is an important diarrhea determinant and vice versa, as undernutrition is associated with a greater incidence and duration of diarrhea, and undernutrition is worsened by diarrhea [18,19,20,21].

Among factors associated with undernutrition and diarrhea, inadequate healthcare and a lack of health services are described in the literature [2,22,23]. These factors result in undernutrition and/or diarrhea being improperly diagnosed and treated, which increases the risk of clinical complications and as a consequence, increases the costs to the health system [23,24]. In Mozambique, most of the healthcare is delivered by the public sector. Service delivery is one of the key building blocks for improving health and strengthening the health system according to the World Health Organization’s (WHO) systems framework, and this has been highlighted since the 2000s [2,25]. A waste of resources is observed when health services are delivered without quality, which is also unethical [1]. The quality of healthcare includes the health workforce, service delivery, financing, leadership and governance, information, the population and their health needs and medical products, vaccines, and technologies [1,2]. Donabedian describes how the quality of care can be classified and assessed under three domains: structure, processes, and outcomes [26].

In LMIC, health services are still lacking desired qualities [1,2,27]. Assessing the availability and readiness of health services allows for the verification of a critical health service’s quality dimension [28]. The assessment of service availability is based on the direct verification of the standards of healthcare, the existence of human resources updated to apply the standards, as well as the existence of equipment, diagnostic capacity, and tracer medicines [1,26,29]. Health service readiness refers to a proper combination of human resources, updated standards of care, and the existence of standards, equipment, diagnostic capacity, and tracer medicines [29]. Service assessments are important for accountability and to inform the improvement of services and are also important for guiding the design and implementation of strategies to improve the quality of service [1,30]. Billah et al. conducted a study on nutrition service assessment in Bangladesh, finding that equipment and training domains had lower scores and could increase nutrition service delivery [29]. Another study conducted in Mozambique in 2000/2001 highlighted that there was room to improve nutrition service delivery for women and children [31]. HIV and malaria are primary healthcare programs that are the highest strategic priorities for the Ministry of Health due to the high burden of these diseases [4,6,13]. HIV and malaria programs benefit from significant specific global funding for service strengthening [32,33].

Few studies in LMIC assess the readiness of nutrition services and diarrhea services for children under five, as well as the influence of malaria and HIV service readiness on nutrition and diarrhea service readiness. This study assesses the structural availability and readiness of health facilities to provide nutrition and diarrhea services for children with undernutrition and/or diarrhea.

## 2. Materials and Methods

### 2.1. Data Source and Collection

The present analysis used data from two studies, i.e., study 1, a secondary analysis of the Service Availability and Readiness Assessment (SARA) implemented in 2018 in Mozambique, and study 2, a field assessment adapted from the Donabedian model and focused on structural readiness, named, nutrition service assessment (NSA), which was a cross-sectional study carried out in three provinces of Mozambique in 2021. SARA 2018 is a representative cross-sectional survey, the NSA 2021 was an assessment implemented in selected sites which were also covered by SARA 2018.

#### 2.1.1. Service Availability and Readiness Assessment (SARA), 2018

The 2018 SARA in Mozambique applied a structured data collection tool to comprehensively verify, at each included health facility, the existence of healthcare standards, human resources, equipment, diagnostic capacity, and tracer medicines of general and specific services, including nutrition services. Mozambique’s SARA data collection tool was designed based on the WHO SARA questionnaire and was validated prior to survey implementation [30]. Data collection was performed using the CSPro application installed on tablets [34]. Details on the SARA methodology have been described elsewhere [30,35].

#### 2.1.2. Nutrition Service Assessment (NSA) in Three Provinces of Mozambique, 2021

Interviews were administered to all health professionals available at the health facilities during the assessment period who were responsible for the management of undernutrition in children under five years of age who were admitted to the hospitals with undernutrition and/or diarrhea. This structured data collection allowed us to assess the availability and readiness of human resources, materials, and infrastructure. The readiness of nutrition services was assessed using a framework adapted from the Donabedian model (Table 1) [36,37].

The checklist and two module questionnaires were designed to capture the essential human power, materials, and infrastructure needed to manage undernutrition according to Mozambique’s standards, as well as international standards, using SARA tools [30,34,38].

Data were collected at five health facilities. The first questionnaire module was administered to health workers and collected information on the academic degree, years of working experience, competency refreshment, and supervision received. Another questionnaire module collected information on the health facility’s human power in terms of numbers and duty profiles, as well as material and equipment availability to manage undernutrition cases, as described in Table 1.

Questionnaires were programmed electronically in Open Data Kit, ODK, version 1.30.1 (University of Washington, Seattle, WA, USA) and administered by the study staff using tablets. Data were stored in ODK Collect and synchronized to the Instituto Nacional de Saúde (INS) ODK Aggregate server.

### 2.2. Study Design and Population

#### 2.2.1. Service Availability and Readiness Assessment—SARA, 2018

Mozambique’s 2018 SARA followed a census approach of public health facilities countrywide, which included 1644 health facilities [34]. The inventory covered health units, namely health posts, health centers, hospitals, other infrastructures of interest to the health sector such as public pharmacies, laboratories, warehouses for medicines and medical commodities, health science institutes, research institutes or equivalents, and other training institutes for health professionals [30,34].

#### 2.2.2. Nutrition Service Assessment—NSA, 2021

A descriptive cross-sectional study was conducted in April 2021 in five health facilities, namely Hospital Central de Maputo (HCM), Hospital Geral José Macamo (HGJM), and Hospital Geral de Mavalane (HGM) in the southern region of Mozambique, Hospital Geral de Quelimane (HGQ) in the central region, and Hospital Central de Nampula (HCN) in the northern region. These are quaternary and tertiary-level healthcare facilities and were purposely selected for the study based on the level of services provided in terms of inpatient care.

### 2.3. Data Analysis

The SARA data allowed us to assess the availability and readiness of nutrition, diarrhea, malaria, and HIV services for children under five. We also analyzed the relationship between the readiness of these four services, specifically whether malaria and HIV service readiness interacts with nutrition and diarrhea service readiness. For the 2021 NSA conducted in three provinces, only the availability and readiness of nutrition services for children were specifically assessed. Readiness scores were defined as the unweighted mean of the availability of human power, guidelines, commodities, diagnosis, and tracer medicine items per service.

For the present analysis, the readiness score was categorized as low (<75%), intermediate (75 to 99%), and ready (100%), based on the relative frequency of the unweighted score capturing the availability of tracer commodities, guidelines, human power, diagnosis, and medicine items [39].

### 2.4. Statistical Analysis

First, descriptive statistics were used to report the availability of tracer items, using absolute and relative frequencies. The readiness percentage score for both datasets was computed, based on the availability score of tracer items for each service, namely undernutrition, diarrhea, HIV, and malaria. Readiness scores were further standardized on a scale from zero to one. The availability scores per services’ tracer items are displayed in Table 2.

Additionally, for the 2018 SARA dataset, we applied second (median) and third quantile regression models to analyse the association between independent variables (e.g., rural/urban areas; facilities with or without inpatient services; readiness scores for HIV and malaria) with the readiness scores for undernutrition and diarrhea services. Simple and multiple quantile models were chosen because the assumptions to perform linear regression models were not satisfied by our data [40]. Friedman’s two-way analysis of variance by ranks was used to compare distributions of readiness by service type, stratified by areas and inpatient services. Post-hoc multiple comparisons were applied when a significant difference was seen. Readiness was summarized in median and interquartile intervals (IQ), i.e., Q1–Q3. We used R version 4.1.0 (Vienna, Austria) to conduct descriptive statistics and the Statistical Package for Social Sciences (SPSS) software version 26.0 to conduct the inferential analysis. *p*-values lower than 5% were considered statistically significant.

## 3. Results

### 3.1. Service Availability and Readiness Assessment—SARA, 2018

For the present analysis, a total of 1644 health facilities were covered in the Mozambique SARA 2018 database, of which 1561 (95.0%) provided services to children under five years of age. From the 1561 health facilities included in the analysis, 85.8% (1339/1561) were in urban areas and 14.0% (218/1561) had inpatient services.

#### 3.1.1. Availability and Readiness

The overall service availability score was 68.6% for nutrition, 73.0% for diarrhea, 72.3% for malaria, and 58.0% for HIV (Table 2). Only 1.2% (15/1295) of the health facilities had all tracer items available for diarrhea, and none presented all tracer items for nutrition services. For malaria, 4.0% (48/1214) of the health facilities had all tracer items available and 0.6% (8/1301) of health facilities had all HIV service tracer items available (Table 2).

The percentage score of service readiness differed depending on the service; it was highest for malaria (median (IQ): 75.0 (66.7 to 83.3)), followed by diarrhea (median (IQ): 72.2 (66.7 to 83.3)), nutrition (median (IQ): 57.1 (52.4 to 57.1)) and then HIV (median (IQ): 55.6 (44.4 to 66.7)), at *p* < 0.001, with malaria being considered intermediate and all other services considered low (Figure 1).

For nutrition services, 49.9% (646/1295) of the health facilities had readiness scores equal to or higher than the mean (58.4%), and higher availability was observed for the dimension of diagnosis capacity (mean = 94.4%). For diarrhea services, 48.3% (625/1295) of the health facilities had a readiness equal to or higher than the mean (74.1%), and higher availability was observed for the equipment dimension (mean = 97.3%). Concerning malaria services, 59.1% (718/1214) of the health facilities had a readiness score equal to or higher than the mean (73.5%), and higher availability was observed for equipment (mean = 98.8%). For HIV services, 48.3% (628/1301) of the health facilities had a readiness score equal to or higher than the mean (60.2%), and higher availability was observed for the equipment dimension (98.8%), but only one item was considered a tracer for this latter assessment.

Figure A1 in Appendix A illustrates the differences in service readiness scores between rural and urban areas. The readiness percentage score differed depending on the services in rural areas and was highest for malaria (median (IQ): 75.0 (66.7 to 83.3)), followed by diarrhea (median (IQ): 72.2 (66.7 to 83.3)), nutrition (median (IQ): 57.1 (52.4 to 66.7)), and HIV services, which presented the lowest readiness percentage score (median (IQ): 55.6 (44.4 to 66.7)) at *p* < 0.001. Only malaria services were considered intermediate, whereas the other services were considered low. The readiness score also differed depending on the services in urban areas, and was highest for malaria (median (IQ): 83.3 (75.0 to 91.7)), followed by diarrhea (median (IQ): 77.8 (72.2 to 84.7)), HIV (median (IQ): 66.7 (55.6 to 77.8)), and nutrition, which presented the lowest readiness score (median (IQ): 61.9 (52.4 to 66.7)) at *p* < 0.001. Malaria and diarrhea were considered intermediate while diarrhea, nutrition and HIV services had low readiness. Details on the descriptive statistics are presented in Table A1, Appendix C. No statistically significant difference was observed in the comparison of diarrhea and malaria service readiness (*p* = 1.000) in health facilities from urban areas.

Figure A2, in Appendix B, presents the differences in the service readiness score according to the health facilities that provide or do not provide inpatient services. Among health facilities with inpatient services, the readiness percentage score was highest for malaria (median (IQ): 91.7 (83.3 to 91.7)), followed by diarrhea (median (IQ): 83.3 (77.8 to 94.4)), HIV (median (IQ): 77.8 (66.7 to 88.9)), and nutrition, which presented the lowest readiness score (median (IQ): 61.9 (57.1 to 66.7)) at *p* < 0.001. Only nutrition services had low readiness, while all other services were considered intermediate. There were no statistically significant differences between diarrhea and malaria service readiness (*p* = 0.163). Among health facilities without inpatient services, the service readiness percentage score was highest for malaria (median (IQ): 75.0 (66.7 to 83.3)), followed by diarrhea (median (IQ): 72.2 (66.7 to 83.3)), nutrition (median (IQ): 57.1 (52.4 to 66.7)), and HIV services, which presented the lowest readiness score (median (IQ): 55.6 (44.4 to 66.7)) at *p* < 0.001. Malaria services were considered intermediate, and other services presented low readiness; details on the descriptive statistics are presented in Table A1, Appendix C. No statistically significant difference was observed in the readiness scores of nutrition and HIV services (*p* = 0.254) within health facilities without inpatient services.

#### 3.1.2. Quantile Regression Model Estimates for Readiness

Table 3 presents the median quantile regression model estimates for the readiness of diarrhea and nutrition services. Health facilities without inpatient services were less likely to score above the median quantile readiness for diarrhea compared to health facilities with inpatient services (adjusted odds ratio: aOR = 0.965, 95% CI: 0.951–0.980; *p* < 0.001). The readiness score for undernutrition, malaria, and HIV services had a positive effect on the median readiness score for diarrhea services, and the highest effect was seen in the undernutrition readiness score (aOR: 2.597, 95% CI: 2.447–2.758; *p* < 0.001), followed by the malaria readiness score (aOR: 1.199, 95% CI: 1.146–1.255; *p* < 0.001), and the lowest effect was seen on the HIV readiness score (aOR: 1.047, 95%: 1.001–1.094; *p* = 0.045).

Health facilities without inpatient services were more likely to score above the median quantile readiness score for undernutrition compared to health facilities with inpatient services (adjusted odds ratio: aOR = 1.029, 95% CI: 1.018–1.041; *p* < 0.001). The readiness score for diarrhea and HIV had a positive effect on the median readiness score for undernutrition (*p* < 0.001). The highest effect was seen on the diarrhea readiness score (aOR: 1.985, 95% CI: 1.911–2.062) and the lowest effect was seen on the HIV readiness score (aOR: 1.098, 95%: 1.054–1.126). The malaria readiness score had a negative effect on the median quantile readiness score for undernutrition (aOR: 0.892, 95% CI: 0.861–0.924).

Third quartile regression estimates for diarrhea and undernutrition are presented in Table A2, Appendix D.

### 3.2. Nutrition Service Assessment: NSA, 2021

From the five health facilities (HF) assessed, there were 67 health professionals, the majority of whom were from HF2 (*n* = 24) and HF5 (*n* = 14). Half of the included professionals were nurses (49.3%; 33/67). The median number of years of work ranged from 2.5 to 8.0 (Table 4).

All wards visited provided diagnostic/treatment services for undernutrition. The guidelines for the integrated management of childhood illness were observed in only one ward (12.5%; 1/8). Six out of eight wards provided growth monitoring services, although only three of the six had growth monitoring guidelines (Figure A3, Appendix E).

The percentage of health professionals who received at least one training on nutritional services in the last two years was 58.2% (39/67), with training on the feeding of babies and young children being the most reported (40.3%; 27/67), and training on the integrated management for childhood illness being the least reported (19.4%; 13/67), as shown in Figure A4a in Appendix F. Specific trainings on nutrition were reported by pediatricians (100%; 6/6), who represent a small fraction of the health professionals linked to nutrition management. Nutritionists were least likely to report with received nutritional training in the previous two years (33.3%; 4/12) (Figure A4b in Appendix F).

Supervision was reported by 53.7% (36/67) of health professionals, whereas 38.8% (26/67) did not receive any supervision related to nutrition services. Of the professionals who received supervision, 19.4% (7/36) received supervision every three months, 22.2% (8/36) received supervision every six months, and 41.7% (15/36) did not specify regularity, as shown in Figure A5a in Appendix G. Most supervision involved nutritionists, at 75.0% (9/12) (Figure A5b in Appendix G).

The overall relative availability of services was 67.5%, with higher availability being observed for the human resources dimension (mean = 87.5%) (Figure A3). None of the health facilities presented all tracer items for nutrition services, as shown in Figure A3 in Appendix E.

The overall readiness score for nutritional services for children was 67.5%. The highest readiness score was at HF2 in the new-born ward, scoring 83.3%, and the lowest nutrition readiness score was observed in the diarrhea ward from HF2 with 53.3%, as depicted in Figure 2.

## 4. Discussion

Our results based on the Service Availability and Readiness Survey, SARA, largely cover the primary healthcare facilities, which represent the majority of national health service facilities, and the focus was on undernutrition, diarrhea, malaria, and HIV services. Nutrition health services were the least available compared to services for diarrhea; the HIV services had the lowest availability among all four assessed services, and diarrhea services had higher availability. Less than 5% of the health facilities had all service tracer items available. None of the health facilities had all of the nutrition service tracer items available. Only 1.2% and 0.6% of the health facilities had all tracer items for diarrhea and HIV services, respectively. The percentage of health facilities with all items is reported as low in other LMIC, where resources are scarce, and the health systems tend to unevenly direct resources to services [1,41,42,43]. Regarding the specific domains, human resources, their training, and guidelines presented lower availability; additionally, for HIV services, the medicine domain also presented lower availability. Our findings on HIV service availability are similar to the availability reported by a Nepalese facility survey conducted in 2015 and from a Tanzanian service provision assessment from 2014 to 2015 [44,45]. Poor human resources availability has been frequently reported in LMIC, where scarce resources and a low percentage of qualified human resources are common features [46]. The lower availability and readiness observed imply that a significant number of children under five are not well diagnosed and do not receive proper health assistance when they pursue these four services with special reference to nutrition and HIV services. A lack of resources in the health services can lead to unresponsiveness to population health needs [1,2,25].

Considering data from the nutrition service assessment conducted in 2021 (NSA, 2021), the overall nutrition services availability was relatively low at 67.5%, while the lowest availability was observed in the guideline domain. None of the included health facilities in the assessment had all tracer items available. The lower service availability that we noticed in 2021 is consistent with the 2018 findings from SARA and is also consistent with findings from other studies conducted in LMIC [29,47,48].

The overall nutrition and diarrhea service readiness scores were low, and for both services, fewer than 50% of the health facilities had a readiness equal to or higher than the entire sample mean readiness. The readiness score of diarrhea services was higher compared to nutrition services. The readiness of malaria services was intermediate; even so, malaria readiness was the highest among the four targeted services, whereas HIV service readiness was the lowest. The historically high burden of malaria and diarrhea in children under five is a possible reason for the malaria service readiness score being the highest and diarrhea services presenting higher readiness than nutrition services [4,49]. The high burden of malaria and diarrhea might therefore explain the high prioritization given to services dealing with high-burden diseases, as part of efforts to reduce child mortality [10,49]. Mozambique has been implementing strategies to reduce the malaria burden, through the Malaria National Control Program from the Ministry of Health, and through different non-governmental organizations [50,51,52,53]. The strategies include improved diagnosis, case management, and availability of medicines [51,52]. The lower service readiness observed for HIV can be explained by the well-documented, unsuccessful results observed in vertical disease programming that results in vertical funding, which is the type of funding observed in the Mozambican health system, especially for HIV [50,54,55,56].

Considering the nutrition service readiness assessment conducted in 2021 (NSA, 2021), the results pointed to hospitals’ diarrhea wards scoring lowest on readiness for undernutrition services. Higher readiness for nutrition services was observed in nutrition-dedicated wards and in new-born wards. Based on the bidirectional relationship observed between nutrition and diarrhea in children under five, the ideal was for both services to be considered ready for nutrition services, or with readiness percentage scores that were approximately the same [18,19]. Better nutrition service readiness in diarrhea wards would allow every child with diarrhea to be assessed for undernutrition and avoid clinical complications and increased costs to the health system [23,24,57].

Nutrition refreshment trainings were frequently provided to pediatricians, of whom there were the fewest in number among those responsible for providing healthcare for undernutrition. However, the 2011–2020 multisectoral action plan for the reduction in chronic undernutrition in Mozambique highlights one of the interventions in the form of nutrition refreshment trainings to nurses [58]. Nutritionists were the health professionals who reported the lowest number of nutrition refreshment trainings. Training on the integrated management of childhood illness was the least frequently provided training to the health workers overall. Regular skills refreshments are well-known strategies for improving service delivery when targeting the technical quality of human resources [1]. Therefore, our result highlights a gap in nutrition service-specific training, and a limited coverage of health worker cadres benefiting from training programs. This scenario might underline the poor technical quality of the services provided [2,30,45,59,60].

The lower integrated management of childhood illness training provision to health workers highlights another key finding from our analysis and substantiates the low availability of guidelines on the integrated management of childhood illness at the point of care. Guidelines for the integrated management of childhood illness are critical resources for scaling up the impact of curative child services, and their implementation is highly recommended to be strengthened within health services [61,62,63,64]. Although supervision was not performed regularly, most supervision targeted nutritionists, which is a positive aspect identified in the study; other authors report that supervision is among the key interventions for improving the quality of services [44,59,61,63,65]. Kruk et al., 2018, highlighted that the combination of training and supervision has a great impact on the better quality of health services [1].

According to the 2018 SARA analysis outputs, diarrhea services had higher readiness than nutrition services, and malaria services had the highest readiness among all analyzed services. Services for HIV had the lowest readiness in rural areas whereas nutrition services had the lowest readiness in urban areas. Nutrition services had the lowest readiness in health facilities offering inpatient services, and HIV services had the lowest readiness in health facilities without inpatient services. Overall, higher service readiness was observed in urban areas and at health facilities with inpatient services. Tertiary and quaternary-level health facilities are mostly located in urban areas and offer inpatient services alongside their specialized services [51]. This could explain the results from this analysis, as specialized services are expected to present better readiness for the services provided.

We aimed to assess the comprehensiveness of readiness in services that are key for child health needs and whether the readiness of one service is associated with the readiness of others. Some services are expected to have better commodities and technical quality due to priority given through verticalized development aid funding toward specific services, which are often justified by the need to reduce the high burden of diseases dealt with by the same services. Services related to malaria and HIV, two health conditions with high prevalence, morbidity, and mortality rates in LMIC, receive priority attention and investments from verticalized global and local health initiatives [4,6,13,66]. In Mozambique, this funding is largely supported by the President’s Emergency Plan for AIDS Relief (PEPFAR), the Global Fund to Fight AIDS, Malaria, and TB, the President’s Malaria Initiative (PMI), and the World Bank [32,33,50]. Even so, some vertical investments may spill over and comprehensively impact child healthcare services since equipment, medicines, human resources, and commodities are shared within and across services. It is therefore expected that the strengthening of some vertical services, which are better funded, might influence the quality of other services targeting children under five. Diarrhea service readiness has a positive association with nutrition, malaria, and HIV service readiness. A higher readiness of diarrhea services is associated with a higher readiness of nutrition services. In turn, nutrition service readiness has a positive association with the provision of inpatient services and with the readiness of diarrhea and HIV services. Nutrition service readiness has the strongest association with diarrhea service readiness. However, at health facilities without inpatient services, which is characteristic of the primary healthcare facilities, when nutrition service readiness increases, the readiness of diarrhea services reduces. These results need to be interpreted carefully, and further analysis is needed to better understand the determinants of these analytical findings. When the readiness of malaria services improves, the readiness of nutrition services decreases. These findings, on the association between the readiness of nutrition and diarrhea services and on the association between the readiness of malaria and nutrition services, suggest the uneven strengthening of services at primary healthcare services in Mozambique, and the lost opportunities for comprehensively supporting health programs’ performance. The finding also suggests that strengthening and maintaining the services available are driven by a verticalized focus on specific diseases rather than system strengthening as a strategy.

None of the health services assessed using either the 2018 SARA or the 2021 NSA can be considered ready if the readiness reference score is set at 100%. Several authors [1,27,29,43,44,67] referred to similar gaps in service readiness and have highlighted unready services as being a widespread situation in LMIC, resulting in poor services provided to address population health needs in these countries [1,25].

The 2018 SARA was a cross-sectional survey. Thus, the assessment did not capture readiness across time, which can be variable depending on logistics, which determine the availability of medical commodities at health facilities. However, the SARA methodology is a reference for situational analyses and policy decisions beyond the timeframe of the cross-sectional study. SARA relied on the verification of tracer items, including medical commodities, medicines, and administrative documents. Data collection went beyond dimensions that are only determined by logistics, but rather are determined by overall health services management and resource availability and allocation. Although the employed verification of items is a cumbersome procedure, which may cause variability in data capturing, SARA surveys were standardized by training, the piloting of study procedures, and the supervision of real-time data collection. Moreover, a sample of the facilities was re-surveyed by a reference team for quality control within one to two weeks after primary data collection. In turn, the 2021 NSA survey was not only a cross-sectional study that was limited in its geographical scope, but it also relied largely on respondents with limited verification processes. Therefore, recall bias may be a limitation, especially for questions about training and supervision. For example, the survey faced a lack of systematic registration or reports on trainings to confirm the answers given by health professionals. Even so, considering that the 2021 evaluation targeted tertiary and quaternary health facilities, and despite the fact that several health facilities sampled do not allow generalization, we still found that specialized services were also unready, which can be indicative of lower readiness across entire health services.

## 5. Conclusions

The availability and readiness of nutrition and diarrhea services for children are low in Mozambique, both scoring below 75%. The unavailability of human resources and their refreshment trainings and guidelines drove these overall lower scores. Diarrhea services had higher readiness compared to nutrition services. A positive association was found between the readiness of nutrition and diarrhea services. HIV service readiness was positively associated with diarrhea and nutrition services. The readiness of malaria services was positively associated with the readiness of diarrhea services; however, it was negatively associated with the readiness of nutrition services. The readiness of nutrition services was lower in urban areas. These results highlight uneven readiness across services that are supposedly equally relevant for primary health care. As expected, the readiness of nutrition services was higher in dedicated nutrition wards and undesirably lower in dedicated diarrhea wards. The situations concerning the readiness of nutrition services in 2018 and 2021 were aligned; however, they scored low for both timepoints, below 75%. Our results also highlight the persisting need to invest comprehensively in service readiness, within and across child health services. In addition, specific trainings, supervision, and the implementation of point-of-care guidelines are critical for improving undernutrition and diarrhea management in Mozambican children. These results can be used as a baseline for monitoring the improvement of the readiness of nutrition, diarrhea, malaria, and HIV services.

## Figures and Tables

**Figure 1 healthcare-10-01200-f001:**
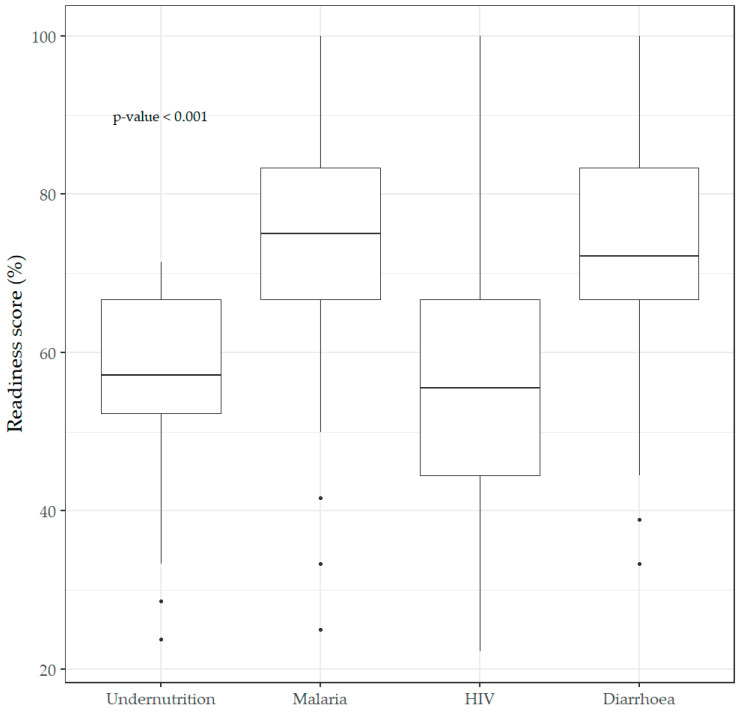
Readiness scores for nutrition, malaria, HIV, and diarrhea services, represented in box plots.

**Figure 2 healthcare-10-01200-f002:**
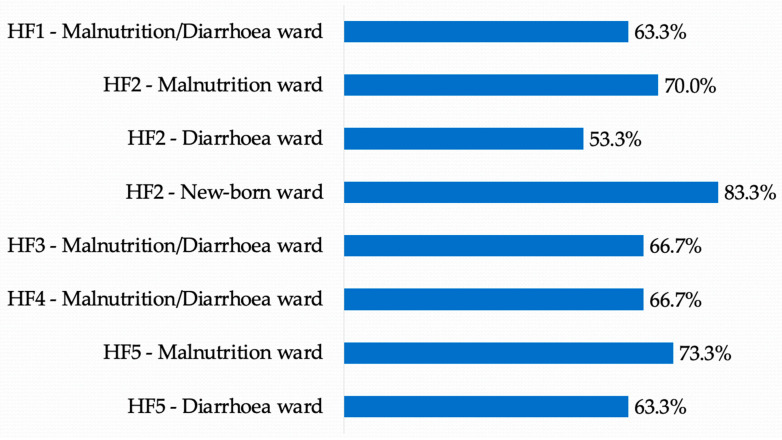
Readiness score for nutrition services, NSA (2021).

**Table 1 healthcare-10-01200-t001:** Structural readiness of the nutrition health service assessment (adapted from the Donabedian model).

Pillar	Indicators	Data Collection Method	Respondents
**Structural Readiness**	Human resources, Infrastructure (by interview and observation), Equipment (including observation), Screening for malnutrition in children with diarrhea as a routine, Clinical meetings to discuss child health management, Guidelines.	Structured questionnaire including a checklist Observation of the existence of functional equipment and materials.	Nurse in charge, nutritionist, and pharmacist.
	Academic profile of health professionals, Experience, Training in the last two years, Supervisions.	Interview using a structured questionnaire.	All health professionals in the pediatric wards who are responsible for the care and monitoring of malnourished children were interviewed.

**Table 2 healthcare-10-01200-t002:** Availability of items for diarrhea, nutrition, malaria, and HIV services. Data source: SARA 2018.

Variables	Diarrhea % (*n*/N)	Nutrition % (*n*/N)	Malaria % (*n*/N)	HIV % (*n*/N)
**Guidelines**				
Guidelines for the integrated care of childhood illness (IMCI)	55.5 (867/1561)	55.5 (867/1561)	-	55.5 (867/1561)
Guidelines on growth monitoring	53.1 (829/1561)	53.1 (829/1561)	-	-
IMCI checklist	-	55.9 (873/1561)	-	-
Malaria guidelines for diagnostics and treatment	-	-	60.3 (939/1556)	-
**Equipment**				
Stadiometer/Altimeter	100.0 (1423/1423)	100.0 (1423/1423)	-	-
Growth curve form	-	58.7 (916/1561)	-	-
Pediatric scale	98.8 (1448/1466)	98.8 (1448/1466)	98.8 (1448/1466)	98.8 (1448/1466)
MUAC tape	93.2 (1455/1561)	93.2 (1403/1498)	-	-
Thermometer	-	-	98.8 (1328/1344)	-
**Differential diagnosis**				
HIV rapid test	95.1 (1425/1498)	95.1 (1425/1498)	-	95.1 (1425/1498)
Malaria rapid test	93.7 (1403/1498)	93.7 (1403/1498)	-	-
Malaria rapid test or microscopy	-	-	97.6 (1462/1498)	-
Blood collection on filter paper	-	-	-	71.3 (1068/1498)
**Medicine/supplies**				
Oral rehydration salt	97.6 (1492/1528)	97.6 (1492/1528)	-	-
Zinc sulfate	41.5 (634/1528)	41.5 (634/1528)	-	-
Vitamin A	91.5 (1398/1528)	91.5 (1398/1528)	-	-
Amoxicillin	85.7 (1309/1528)	85.7 (1309/1528)	-	-
Cotrimoxazole	88.1 (1346/1528)	88.1 (1346/1528)	-	-
Albendazole/Metronidazole *	91.0 (1390/1528)	97.2 (1485/1528)	-	-
Coarten (artemeter and lumefantrina)	-	-	93.3 (1421/1523)	-
Injectable or rectal artesunate	-	-	58.8 (896/1523)	-
Quinine	-	-	62.4 (951/1523)	-
Paracetamol or Ibuprofen	-	-	88.9 (1359/1528)	-
Nevirapine syrup (NVP) or Lopinavir (LVP) + Ritonavir (RTV)	-	-	-	95.6 (821/859)
Zitovudina (AZT) or Abacavir (ABC)	-	-	-	42.9 (600/1399)
Lamivudine (3TC)	-	-	-	7.4 (104/1399)
**Human resources and training**				
Training on IMCI in the last two years	46.1 (720/1561)	46.1 (720/1561)		
Training in growth monitoring services in the last two years	35.0 (546/1561)	35.0 (546/1561)	-	-
General clinicians	7.2 (113/1561)	7.2 (113/1561)	7.2 (113/1561)	7.2 (113/1561)
General nurses	50.9 (795/1561)	50.9 (795/1561)	59.2 (924/1561)	-
Nutritionists	-	8.8 (117/1330)	-	-
Maternal and child health nurses	87.4 (1365/1561)	87.4 (1365/1561)	-	87.4 (1365/1561)
Medical technicians	-	-	55.1 (860/1561)	55.1 (860/1561)
**Overall items’ relative availability**	73.0	68.6	72.3	58.0
**Health facilities with all items**	1.2 (15/1295)	0.0 (0/1295)	4.0 (48/1214)	0.6 (8/1301)

* Albendazole’s availability was only evaluated for nutrition services; *n*: number of units in the subgroup; N: total number of units.

**Table 3 healthcare-10-01200-t003:** Median quantile regression model estimates for diarrhea and undernutrition readiness.

Variables	*p*-Value	cOR ^1^ (CI ^2^ 95%)	*p*-Value	aOR ^3^ (CI ^2^ 95%)
Median quantile regression model estimates for diarrhea readiness				
Type of area	<0.001		0.495	
*Urban*		1		1
*Rural*		0.946 (0.919–0.974)		0.995 (0.981–1.010)
Type of health facility	<0.001		<0.001	
*With inpatient services*		1		1
*Without inpatient services*		0.895 (0.869–0.921)		0.965 (0.951–0.980)
Readiness for undernutrition	<0.001	3.211 (3.046–3.385)	<0.001	2.597 (2.447–2.758)
Readiness for malaria	<0.001	1.560 (1.446–1.683)	<0.001	1.199 (1.146–1.255)
Readiness for HIV	<0.001	1.649 (1.542–1.763)	0.045	1.047 (1.001–1.094)
Median quantile regression model estimates for undernutrition readiness				
Type of area	<0.001		0.087	
*Urban*		1		1
*Rural*		0.953 (0.930–0.977)		1.010 (0.999–1.021)
Type of health facility	<0.001		<0.001	
*With inpatient services*		1		1
*Without inpatient services*		0.953 (0.930–0.977)		1.029 (1.018–1.041)
Readiness for diarrhea	<0.001	1.902 (1.837–1.970)	<0.001	1.985 (1.911–2.062)
Readiness for malaria	<0.001	1.331 (1.247–1.420)	<0.001	0.892 (0.861–0.924)
Readiness for HIV	<0.001	1.535 (1.449–1.626)	<0.001	1.089 (1.054–1.126)

^1^ Unadjusted odds ratio; ^2^ 95% confidence intervals; ^3^ adjusted odds ratio.

**Table 4 healthcare-10-01200-t004:** Distribution of health professionals by health facilities (HF), roles, and years of work (2021).

	*n* (%)
**Number of health professionals by health facility (HF)**	
HF2	24 (35.8)
HF5	14 (20.9)
HF1	12 (17.9)
HF4	9 (13.4)
HF3	8 (11.9)
**Roles of the health professionals**	
Nurse	33 (49.3)
Generalist medical doctor	15 (22.4)
Nutritionist	12 (17.9)
Pediatrician	6 (9.0)
Technician (preventive medicine)	1 (1.5)
**Years of work by HF**	**Median (IQ), min–max**
HF2	3.0 (1.75–7.25), 0–30
HF5	2.5 (2.0–6.5), 0–16
HF1	3.5 (1.0–6.75), 0–12
HF4	8.0 (3.0–15.0), 0–16
HF3	3.5 (1.5–5.25), 0–20

*n*: number of units in the subgroup.

## Data Availability

The data from the 2018 Service Availability and Readiness Assessment are available upon reasonable request by e-mail to Instituto Nacional de Saúde from Mozambique or the World Health Organization’s representation in Mozambique, or the Directorate of Planning and Cooperation at Mozambique’s Ministry of Health. The data are subject to a data-sharing agreement. The 2021 nutrition service assessment data are not publicly available due to restrictions present in the consent forms. The data presented in this study are available from the corresponding author upon reasonable request subject to a data-sharing agreement.

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
