# Peer review of "Readiness of Mozambique Health Facilities to Address Undernutrition and Diarrhea in Children under Five: Indicators from 2018 and 2021 Survey Data"

_healthcare, 2022, doi:10.3390/healthcare10071200_

Round 1

Reviewer 1 Report

In this study, the authors asesses the structural availability and readiness of health facilities to provide nutrition and diarrhoea services for children with undernutrition and/or diarrhoea. The subject is very important, and is one to which the author has made significant contributions. I do however have a few comments that need to be addressed.

Introduction. I would strongly advise the authors to rewrite their introduction more contextualized to health facilities readiness to address undernutrition and diarrhoea in children 
Methods. Was there a pilot survey? the adaptation of the questionnaire was validated?
Results. The attached figures should be converted into tables and included in the results section.
Discussion. Authors should include and discuss the limitations of their study in a paragraph.

Author Response

In this study, the authors assess the structural availability and readiness of health facilities to provide nutrition and diarrhoea services for children with undernutrition and/or diarrhoea. The subject is very important, and is one to which the author has made significant contributions. I do however have a few comments that need to be addressed.

Introduction. I would strongly advise the authors to rewrite their introduction more contextualized to health facilities readiness to address undernutrition and diarrhoea in children 
Methods. Was there a pilot survey? the adaptation of the questionnaire was validated?
Results. The attached figures should be converted into tables and included in the results section.
Discussion. Authors should include and discuss the limitations of their study in a paragraph.

            Response: We would like to thank for the comments.

Introduction, information was added, lines 294-298.

Methods, a pre-test/validation process was conducted to evaluate the data collection tools and the enumerators capacity for SARA component. This component was added, lines 688-689.

Results, the figures were converted into tables and added to the main results section.

Discussion, the limitations are presented in the last paragraph of the discussion, lines 2697-2717.

Reviewer 2 Report

Due to the numerous segmentation of the content, especially in the method part, the research design seems ambiguous. The authors need to clearly explain what method they have used to answer the research question and what results they have obtained. They should also explain how the questions of questionnaire prepared are in line with the objectives of the study.

Author Response

Due to the numerous segmentation of the content, especially in the method part, the research design seems ambiguous. The authors need to clearly explain what method they have used to answer the research question and what results they have obtained. They should also explain how the questions of questionnaire prepared are in line with the objectives of the study

Response: We would like to thank the reviewer for the comment.

In this paper we present 2 survey data: first a secondary analysis of the Service Availability and Readiness Assessment (SARA), which the methodology is published elsewhere and referred to, and overview mentioned in the proposed article lines 690-691; second, a field assessment adapted from the Donabedian model, and its focused on structural readiness, which we call it NSA, and describe in lines 693-699. Whilst SARA 2018, is a representative cross-sectional survey, the NSA 2021 is an assessment implemented in selected sites which were also covered by SARA. Applied questionnaires are essentially based on SARA methods and standards, these are validated instruments, as mentioned in lines 688-689.

Reviewer 3 Report

Title: Could be improved. Such as:  Readiness of Mozambique healthcare to.... in under-5 children:  indicators from 2018 and 2021 survey data.

Overall the paper lacks clarity in both the abstract and main text. It is obvious 2018 is a representative cross-sectional survey whilst 2021 is a subanalyses of selected sites. The authors are not providing precise descriptive terms to address these questions- What is been studied? Who is selected? How is it evaluated? What are the outputs/indicators?

Abstract:

Lines 22-25- Requires language editing. Too complex.

Lines 33-34- This is a conclusion and not a result output.

Introduction:

Line 38- the term ‘children under-five’ is not grammatically correct, when ‘under-five’ is an adjective. The correct terms is ‘under-five children’. And it is possible to shorten this further progressively to ‘under-5s’.

Line 43-44: Are the authors sure that undernutrition increases the risk of severity of diarrhoea? or is it undernutrition risk develops from chronic/ repeated diarrhoea episodes? It’s the reverse. Further, risk level data should be provided to indicator the extent of the issue.

The last sentence [lines 52-54] is not correct. It is important to distinguish 'association' from 'cause'. The prevalence of undernutrition in X site [19], had co-morbidities noted. But although under-5s from the rural slum area were less undernourished [stunting, wasting etc], diarrhoea was significant  with wasting here compared to the urban under 5s. Was hygiene or water quality the deciding factor for diarrhoea incidence? Now this is a general comment. Is undernutrition the ‘ONLY’ cause of diarrhoea, or others- such as water quality and conditions of supply?

The 4th paragraph clearly is about the study rationale. But the 1st sentence is ambiguous without a rationale. Could the authors introduce the standard metric for ‘desired quality? Expand on this theme about what is quality? How can it be measured? Other countries’ quality measures? The rest of this paragraph is better suited to the discussion, where verticalized development can be blamed for poor quality healthcare.

Methodology:

A clear separation of data sources should be provided for clarity. Essentially the 2018 SARA survey is a nationally representative survey benchmarked to WHO. Whilst the NSA data is a situational analyses of 3 of those sites [confirm this or if not , state they are different]. Clarity of sampling, what indicators are extracted from each sampling year etc must be provided. [and written accordingly]. An algorithm figure will better define this process. If Table 1 is providing indicators for NSA, then there should be a similar data table for SARA data.

The adoption of the readiness scoring [For the present analysis, readiness score was categorized as low (<75%), intermediate 148 (75 to 99%) and ready (100%), based…] is from a Bangladesh study. The geo-political conditions of Mozambique and Bangladesh are vastly different. Have the authors validated this score? Otherwise provide a rationale for this adoption.

Results:

Scoring should be emphasized according to the cutoffs, i.e … readiness score was categorized as low (<75%), intermediate 148 (75 to 99%) and ready (100%)…

Discussion:

Please center this on ‘readiness’. The 1st sentence of the 1st paragraph should be the last sentence of this paragraph or the section.

Scoring should be emphasized according to the cutoffs, i.e … readiness score was categorized as low (<75%), intermediate 148 (75 to 99%) and ready (100%)…

Talk about verticalization of development [transfer from introduction].

Overall discussion should be better crafted and not repeat results. Instead focus on implications of those findings.

Conclusion- should be restricted to key findings and implications.

Statistics, Tables and figures- no issues

Author Response

Note: The manuscript was submitted to English editing us recommended.

Title: Could be improved. Such as:  Readiness of Mozambique healthcare to.... in under-5 children:  indicators from 2018 and 2021 survey data.

Overall, the paper lacks clarity in both the abstract and main text. It is obvious 2018 is a representative cross-sectional survey whilst 2021 is a sub analyses of selected sites. The authors are not providing precise descriptive terms to address these questions- What is been studied? Who is selected? How is it evaluated? What are the outputs/indicators? 

Response: we would like to thank the reviewer for the comments. Regarding the title, it was improved as suggested. Some sections of the document were revised and edited.

Abstract: 

Lines 22-25- Requires language editing. Too complex.

            Response: Thank you for the comment, the sentence was revised, lines 22-28.

Lines 33-34- This is a conclusion and not a result output.

Response: Thank you for the comment, in the referred lines the objective is to present the conclusion, edited lines 34-36.

Introduction:

Line 38- the term ‘children under-five’ is not grammatically correct, when ‘under-five’ is an adjective. The correct terms is ‘under-five children’. And it is possible to shorten this further progressively to ‘under-5s’.

Response: We would like to thank the reviewer, the grammatical error was corrected, line 40. We would also like to thank for the suggestion to shorten the term under-five, but we would rather keep it the way it is presented, to standardize with other publications made by some of the authors listed here and with other publications worldwide.

Line 43-44: Are the authors sure that undernutrition increases the risk of severity of diarrhoea? or is it undernutrition risk develops from chronic/ repeated diarrhoea episodes? It’s the reverse. Further, risk level data should be provided to indicator the extent of the issue.

Response: Thank you for the comment, the relationship between undernutrition and diarrhoea is bidirectional, this relationship is well detailed by Brown, 2003. Tickell et all, 2017 presents the association between wasting and severity of diarrhoea. For example, one of the indicators that could define the severity of diarrhoea is its duration. The association of undernutrition and diarrhoea duration is described by Black et al 1984, reference added to the manuscript.

References:

  1. Brown, K.H. Diarrhea and Malnutrition. The Journal of Nutrition 2003, 133(1), 328s–332s.

  1. Tickell, K.D.; Pavlinac, P.B.; John-Stewart, G.C.; Denno, D.M.; Richardson, B.A.; Naulikha, J.M.; Kirera, R.K.; Swierczewski, B.E.; Singa, B.O.; Walson, J.L. Impact of Childhood Nutritional Status on Pathogen Prevalence and Severity of Acute Diarrhea. The American Journal of Tropical Medicine and Hygiene 2017, 97, 1337–1344, doi:10.4269/ajtmh.17-0139.

  1. Black, R.E.; Brown, K.H.; Becker, S. Malnutrition Is a Determining Factor in Diarrheal Duration, but Not Incidence, among Young Children in a Longitudinal Study in Rural Bangladesh. The American Journal of Clinical Nutrition 1984, 39, 87–94, doi:10.1093/ajcn/39.1.87.

The last sentence [lines 52-54] is not correct. It is important to distinguish 'association' from 'cause'. The prevalence of undernutrition in X site [19], had co-morbidities noted. But although under-5s from the rural slum area were less undernourished [stunting, wasting etc], diarrhoea was significant with wasting here compared to the urban under 5s. Was hygiene or water quality the deciding factor for diarrhoea incidence? Now this is a general comment. Is undernutrition the ‘ONLY’ cause of diarrhoea, or others- such as water quality and conditions of supply?

Response: Thank you for the comment, the relationship between undernutrition and diarrhoea is bidirectional, this relationship is well detailed by Brown, 2003. The term cause was changed to associated. Yes, many other factors are known for causing diarrhoea.

The 4th paragraph clearly is about the study rationale. But the 1st sentence is ambiguous without a rationale. Could the authors introduce the standard metric for ‘desired quality? Expand on this theme about what is quality? How can it be measured? Other countries’ quality measures? The rest of this paragraph is better suited to the discussion, where verticalized development can be blamed for poor quality healthcare.

Response: Thank you for the comment, information about quality of care was added, line 281-284. The information on verticalization was moved to the discussion, lines 2430-2442.

Methodology:

A clear separation of data sources should be provided for clarity. Essentially the 2018 SARA survey is a nationally representative survey benchmarked to WHO. Whilst the NSA data is a situational analysis of 3 of those sites [confirm this or if not, state they are different]. Clarity of sampling, what indicators are extracted from each sampling year etc must be provided. [and written accordingly]. An algorithm figure will better define this process. If Table 1 is providing indicators for NSA, then there should be a similar data table for SARA data.

            Response: We would like to thank for the comment, the variables investigated are now presented in table 2 for SARA.

The adoption of the readiness scoring [For the present analysis, readiness score was categorized as low (<75%), intermediate 148 (75 to 99%) and ready (100%), based…] is from a Bangladesh study. The geo-political conditions of Mozambique and Bangladesh are vastly different. Have the authors validated this score? Otherwise provide a rationale for this adoption.

            Response: There is no globally agreed score for readiness, different authors use different approach, and Mozambique didn’t define readiness scores, so based on the assumption that when analysing the quality of care the health facility should have all tracer items we adopted the 100% as ready. The Organization and governance of Bangladesh health system is almost the same as Mozambique REF. WHO, 2015, link:https://apps.who.int/iris/bitstream/handle/10665/208214/9789290617051_eng.pdf?sequence=1&isAllowed=y , Mozambique health sector strategic plan http://www.africanchildforum.org/clr/policy%20per%20country/2018%20Update/Mozambique/mozambique_healthsectorstrategicplan_2014-2019_en.pdf

Results:

Scoring should be emphasized according to the cutoffs, i.e … readiness score was categorized as low (<75%), intermediate 148 (75 to 99%) and ready (100%).

            Response: We would like to thank for the comment, we emphasized the results based on the cut-offs as suggested.

Discussion:

Please center this on ‘readiness’. The 1st sentence of the 1st paragraph should be the last sentence of this paragraph or the section.

Response: Thank you for the comment, the sentence was moved as suggested, lines 532-533.

Scoring should be emphasized according to the cutoffs, i.e … readiness score was categorized as low (<75%), intermediate 148 (75 to 99%) and ready (100%).

            Response: We would like to thank for the comment, scoring for readiness was emphasized according to the cut-offs as suggested.

Talk about verticalization of development [transfer from introduction].

            Response: Thanks for your comments. Indeed, we moved from introduction, text indicating a scenario of verticalization, so that this is not contextualized in discussion and conclusion, lines 2430-2442.

Overall discussion should be better crafted and not repeat results. Instead focus on implications of those findings.

            Response: We would like to thank for the comment, the discussion was edited.

Conclusion- should be restricted to key findings and implications.

            Response: Thanks for your comments. We revised the conclusion to be based on results and implications strictly related to study results.

Reviewer 4 Report

The authors assessed the readiness score of nutrition and diarrhoea services and the influence of malaria and HIV service readiness on these two services' readiness in public/referral health facilities in Mozambique. Although the topic may be potentially important in this field, there are several concerns, as mentioned below.

1) Introduction

The authors should explain what is already known and unknown about the present theme in this section and state the necessity of the present study. Therefore, the authors should explain why they targeted public/referral health facilities in Mozambique. Does Mozambique have unique features compared to other African countries or LMIC? Or is it considered representative of these countries? I also did not understand why they used both SARA and NSA datasets because the data collection procedures and outcomes differed between these surveys. Additionally, some descriptions should be sufficiently referred to by appropriate rationale. Although the authors mentioned the absence of previous studies , they compared the present results with the previous ones in the discussions section.

2) Methods

Descriptions are insufficient and confusing despite the redundancy. Please clearly describe the variables investigated and the procedures, although it is considered that the authors simply tabulated the frequencies of variables. In Table 1, it is difficult to determine the correspondence with the Donabedian model and understand why the authors provide two rows. Some subtitles used and the paragraphs followed were inappropriate.

3) Results

This section is too long; the authors do not have to mention all results from Tables and Figures. Most results provided in Figures should be provided in Tables. It is a common mistake to split up into several Tables data that belong in one Table.

4) Discussion

Given the inappropriate methodology of this study, it is difficult to evaluate the discussion section. However, the authors should delete the first sentence because this section should be started with a summary of the main results of the present study.

Author Response

The authors assessed the readiness score of nutrition and diarrhoea services and the influence of malaria and HIV service readiness on these two services' readiness in public/referral health facilities in Mozambique. Although the topic may be potentially important in this field, there are several concerns, as mentioned below.

1) Introduction 

The authors should explain what is already known and unknown about the present theme in this section and state the necessity of the present study. Therefore, the authors should explain why they targeted public/referral health facilities in Mozambique. Does Mozambique have unique features compared to other African countries or LMIC? Or is it considered representative of these countries? I also did not understand why they used both SARA and NSA datasets because the data collection procedures and outcomes differed between these surveys. Additionally, some descriptions should be sufficiently referred to by appropriate rationale. Although the authors mentioned the absence of previous studies, they compared the present results with the previous ones in the discussions section.

            Response: Thank you for the comment. The introduction is divided in paragraphs, the first paragraph is related to the burden of the diseases that we assess the services, the second is related to the association between undernutrition and diarrhoea. In the third paragraph we start to discuss what is know about quality of services in LMIC, and the 4 paragraph is about the study rationality. The public health facilities represent the majority of healthcare provision. added line 276. No, Mozambique settings are similar to other LMIC but no data on national service availability and readiness is yet published and additional as described in the last paragraph of the introduction, we are presenting a unique analysis on the association of different services readiness. The ones that we used in the discussion unfortunately were not conducted to analyse the association between undernutrition and diarrhoea for example. The SARA survey was a nationally representative survey benchmarked to WHO and the NSA was a situational/updated analyses of 3 of those sites with more specific indicators they assess the readiness to deliver nutrition services for children under-five with undernutrition and/or diarrhoea.

2) Methods

Descriptions are insufficient and confusing despite the redundancy. Please clearly describe the variables investigated and the procedures, although it is considered that the authors simply tabulated the frequencies of variables. In Table 1, it is difficult to determine the correspondence with the Donabedian model and understand why the authors provide two rows. Some subtitles used and the paragraphs followed were inappropriate.

            Response: We would like to thank for the comment, the variables investigated are now presented in table 2 for SARA and table 1 for NSA, for SARA we decided to directly refer to it while present the frequencies to avoid having two long tables with partial same information, as same indicators vary from service to service. The procedures are described in methods section 2.2.1 and 2.2.2, the detailed procedure for data collection for SARA are described by O’Neill et al, 2013 and WHO, SARA reference manual, these two references are cited in line 211. The Donabedian model accesses the quality of services based on three dimensions: Structural readiness, Process and Outcome, for the NSA we used only the Structural Dimension considering the indicators proposed by this model, but we also added some indicators (all the indicators presented in the second row excluding the training that is already proposed in the Donabedian model). We separated the indicators in two rows because of the data collection method that was different. We added some information about the Donabedian model in the introduction lines 283-285.

3) Results

This section is too long; the authors do not have to mention all results from Tables and Figures. Most results provided in Figures should be provided in Tables. It is a common mistake to split up into several Tables data that belong in one Table.

            Response: Thank you for the comment, we did change the section and the presentation of some results, we summarized the presentation of some results.

4) Discussion

Given the inappropriate methodology of this study, it is difficult to evaluate the discussion section. However, the authors should delete the first sentence because this section should be started with a summary of the main results of the present study.

            Response: We would like to thank for the comment, as suggested the first sentence was removed from the beginning of the discussion.

Round 2

Reviewer 2 Report

Dear Authors,

Please add the following sentences in the method section:

In this paper we present 2 survey data: first a secondary analysis of the Service Availability and Readiness Assessment (SARA), which the methodology is published elsewhere and referred to, and overview mentioned in the proposed article lines 690-691; second, a field assessment adapted from the Donabedian model, and its focused on structural readiness, which we call it NSA, and describe in lines 693-699. Whilst SARA 2018, is a representative cross-sectional survey, the NSA 2021 is an assessment implemented in selected sites which were also covered by SARA. Applied questionnaires are essentially based on SARA methods and standards, these are validated instruments, as mentioned in lines 688-689. 

Author Response

Reviewer #2 Round #2

Please add the following sentences in the method section:

In this paper we present 2 survey data: first a secondary analysis of the Service Availability and Readiness Assessment (SARA), which the methodology is published elsewhere and referred to, and overview mentioned in the proposed article lines 690-691; second, a field assessment adapted from the Donabedian model, and its focused on structural readiness, which we call it NSA, and describe in lines 693-699. Whilst SARA 2018, is a representative cross-sectional survey, the NSA 2021 is an assessment implemented in selected sites which were also covered by SARA. Applied questionnaires are essentially based on SARA methods and standards, these are validated instruments, as mentioned in lines 688-689. 

Response: We would like to thank the reviewer for the comment, the sentence was added to the methods, lines 317-685.

Yours sincerely,

Júlia Sambo, Pharmd, MPH

Instituto Nacional de Saúde, EN1, Bairro da Vila – Parcela n3943, Distrito de Marracuene, Mozambique; E-mail: [email protected]
